# Disruption in daily eating-fasting and activity-rest cycles in Indian adolescents attending school

**Neelu Jain Gupta**⬤*, **Akansha Khare**

Department of Zoology, Chaudhary Charan Singh University, Meerut, UP, India

* drneelujgupta@hotmail.com

**Data Availability Statement:** All relevant data are within the paper.

**Funding:** The author(s) received no specific funds for the study but support in form of Actiwatches for

## Abstract

A lifestyle with erratic eating patterns and habits predisposes youngsters to obesity. Through a two-phase feasibility study among Indian students living in the Delhi area, we longitudinally examined the following: (1) the daily eating-fasting cycles of students (N = 34) in school and college using smartphones as they transition from high school (aged 13–15 years; $n_{IX}$ = 13) to higher secondary school (HSSS; 16–18 years; $n_{XII}$ = 9) to their first year (FY) of college (18–19 years; $n_{FC}$ = 12); and (2) daily activity-rest cycles and light-dark exposure of 31 higher secondary school students (HSSS) using actigraphy. In phase 1, students' food data were analyzed for temporal details of eating events and observable differences in diet composition, such as an energy-dense diet (fast food (FF)), as confounding factors of circadian health. Overall, the mean eating duration in high school, higher secondary and FY college students ranged from 14.1 to 16.2h. HSSS exhibited the shortest night fasting. Although FY college students exhibited the highest fast food percentage (FF%), a positive correlation between body mass index (BMI) and FF% was observed only among HSSS. Furthermore, the body weight of HSSS was significantly higher, indicating that FF, untimely eating and reduced night fasting were important obesity-associated factors in adolescents. Reduced night fasting duration was also related to shorter sleep in HSSS. Therefore, food data were supplemented with wrist actigraphy, i.e., activity-rest data, in HSSS. Actigraphy externally validated the increased obesogenic consequences of deregulated eating rhythms in HSSS. CamNtech motion watches were used to assess the relationship between disturbed activity cycles of HSSS and other circadian clock-related rhythms, such as sleep. Less than 50% of Indian HSSS slept 6 hours or more per night. Seven of 31 students remained awake throughout the night, during which they had more than 20% of their daily light exposure. Three nonparametric circadian rhythm analysis (NPCRA) variables revealed circadian disruption of activity in HSSS. The present study suggests that inappropriate timing and quality of food and sleep disturbances are important determinants of circadian disruptions in adolescents attending school.

the study was provided by Prof. Satchidananda Panda for this work

**Competing interests:** The authors have declared that no competing interests exist

## Introduction

Longitudinal studies on circadian rhythms are needed to support the contemporary prognosis of feeding-fasting and sleep-wake rhythm disruption in adolescents. Lifestyle traits such as faulty food habits and poor sleep are critical risk factors in young adults and can contribute to the development of psychopathological alterations later in life [1]. Studies reveal that adhering to traditional healthy food influences adolescents' daily feeding behavior [2], indicating that a high-calorie diet is associated with meal irregularity. Although food type [3], frequency and circadian timing of meals have synergistic effects on individual health [4] at all ages, adolescent diets and circadian health merit special attention. Biologically, adolescence marks an age when the fundamental properties of the circadian timekeeping system change and develop [5]. Socially, the daily routines of students change as they transition from high school to higher secondary schools [6, 7]. Psychologically, study pressures and reduced school connectedness lower their mental health [8]. We propose that untimely eating and diminished sleep alongside pubertal changes negatively impact the well-being of adolescents attending school.

On the Indian subcontinent, 23.1% [9] of the 243 million adolescents [10] lead urban lives, and 77% of these adolescents attend school. Adolescents study in school at two levels: high school (aged 13–15 years) and higher secondary school (16–18 years). Most of the schools in these age groups have a uniform pattern of school curricula, rendering common study structures and grading systems across the country. Thus, there are some education system commonalities among Indian adolescents, which include less emphasis on practical learning and physical education and the use of marks/exam scores as primary indicators of success. Implicitly, the daily routine commonalities among adolescents attending school include late-night studying, a larger focus on short-term goals, attraction to fast food (FF) and weekday/weekend switches in daily eating and sleeping routines. All these traits lead to circadian disruption and need to be comprehensively investigated. Our earlier study suggesting the disruption of daily eating rhythms [11] in Indian adults could not be extended to students attending school, who might experience a different nature of fluctuations in food routines, more frequent variations in taste(s) for food types, etc. The transition from high school adolescent life to young adult college life witnesses marked changes in access to FF and time(s) of daily food intake [12]. Furthermore, there is no study describing the daily sleep patterns of Indian adolescents who attend school.

Circadian rhythms have a consolidative role in organismal physiology. Studies have shown that molecular circadian clockwork and time of food intake interactively enable temporal re-appropriation of organismal energy budgeting and optimize health [13, 14]. Likewise, eating wisely through the right time of day confers a survival advantage by resetting various cellular and tissue functions to the correct time of day; thus, this eating pattern has the potential to correct circadian misalignment [15]. Intervention studies in mice [16, 17] in which the time of food intake was restricted (hence, time-restricted eating (TRE)) to 8–10 hours a day revealed that TRE had the potential to reverse the predisposition to metabolic disorder. However, mice and humans are different physiologically, genetically and even ecologically [18]. In animal model studies, food intake largely depends on food administered, whereas human food eating behavior has several confounding factors viz. age, busy-ness, eating preferences, urbanization, light exposure, moods, socioeconomic status, and health awareness. Therefore, factual and evidence-driven descriptions of what and when people actually eat must precede human intervention studies [11, 19, 20].

A scientific testing of public notions, such as the influence of caloric diet on circadian disruption, must be tested in light of the cumulative influence of multiple factors (such as reduced sleep, less activity, artificial light at night) on circadian health [21]. For example, the differences

in obesogenic consequences of high caloric food consumption between Indian high school and HSSS [22] could be associated with study-related stress, reduced sleep, activity and even Indian culture [23]. The inverse relationship between sleep duration and weight status has revealed the importance of sleep in nutritional health [24]. In addition to sleep, ambient light regulates human circadian rhythms by regulating melatonin production from the pineal gland and by influencing sleep initiation and maintenance [25]. Cross-sectional and longitudinal actigraphy is a standard method [26] to determine the relationship between disturbed activity cycles and other circadian clock-related rhythms. Therefore, to externally validate the constituent factors augmenting the extent of daily rhythm deregulation [27] in food cycles of HSSS, observations on activity/rest and light exposure data were made using wrist actigraphy. Three nonparametric circadian rhythm analysis (NPCRA) variables helpful in validation of "actimetric sleep" were calculated using motion-watch software tools [28–31]. These characteristics, i.e., relative amplitude (RA), intradaily variability (IV) and interdaily stability (IS), tested the fragmentation and synchronization of the sleep-wake cycle. Herein, the relative amplitude ranges between maximum and minimum levels of activity. Intradaily variability (IV) is a measure of the degree of fragmentation and denotes the frequency and extent of hourly transitions between periods of rest and activity. Interdaily stability (IS) estimates the strength of coupling external cue-triggering activity and indicates synchronization to the 24-h light-dark cycle.

Substantial chronobiological research is dedicated to the mechanisms and genetic bases of biological clocks [32–34]. Fewer studies have identified relevant societal perturbations that contribute to daily rhythm deregulation. These descriptive approaches append the effectiveness of studies targeting an ultimate human benefit. Therefore, a preliminary multifactorial feasibility study was performed in two phases using a combination of 1) daily eating patterns among Indian school students studying in classes IX and XII and FY of college and 2) longitudinal monitoring of diurnal changes in the intensity patterns of the free-living activity and light exposure of higher secondary school students (HSSS), which would help delineate the lifestyle of school students and establish the relative importance of causal factors to circadian disruption.

## Methods

### Inclusion and exclusion criteria for food pattern study

The study protocol was approved by the human ethics committee (HEC) of MMH College, Ghaziabad, India. The inclusion criteria were as follows: (1) students were enrolled in class IX, class XII or FY of college and were aged, respectively, 13–15, 16–18 or 18–19 years at inclusion; (2) their parents agreed that mobile data capturing was not detrimental to studies; and (3) they had continued access to a mobile device (provided by the researcher) at home or school. The exclusion criteria were as follows: (1) undergoing any weight-loss/weight-gain program, having suffered a sickness or disease or having taken appetite-related medication in the past 6 months; (2) having taken any psychoactive drugs in the past two years; (3) having a history of seizures or any condition that prevented them from participating in school activities in the past 2 years; (4) discouraged by class teachers, based on discipline records; (5) undertook travel during the study; (6) stayed in school/college hostel; (7) having special needs; (8) having a medical illness requiring immediate treatment; and (9) having a disinterest/inability to commit to the regularity of food picture capturing. In addition to obtaining students' consent to participate, we also collected parental consent for their child to be part of the study. We also obtained permission from the school administrator for student(s) to participate in the study, as cell phone use is forbidden in many high schools in India.

### A camera phone method to collect eating pattern data

Questionnaire-based methods or food diaries to collect eating behavior often lack the circadian aspect of eating behavior and its variation from day to day. In phase 1, data capturing was performed using mobile phone cameras. Data collection and analysis were performed as published earlier [11]. Briefly, food metadata were collected from study participants attending school who took pictures of every ingestion event for 21 days using the phone handset provided and by study participants attending college using their personal phone handsets. In either case, food pictures were manually downloaded, time-stamped and annotated by researchers.

Initially, in the food eating pattern study, 7 schools were approached, of which 2 schools allowed the researchers to contact students for enrollment (Fig 1). Fifty-six students and parents consented to participate. Of these students, 18 dropped out and 4 were irregular; as such, we obtained the daily eating-fasting cycle data of 34 student volunteers using smart phones, namely, students studying in high school (aged 13–15 years; $n_{IX}$ = 13), higher secondary school (aged 16–18 years; $n_{XII}$ = 9; HSSS), and their first year (FY) of college (aged 18–19 years; $n_{FC}$ = 12). Ten functional camera phone handsets (Nokia C1 with data storage but without server connectivity) with 2GB memory cards were given to recruited students after pre-screening and consent. They were asked to record all of their food and beverage (including water) intake, irrespective of serving size, and save the pictures in their phone. The students from one college were contacted via flyers, who after consent, were directly recruited by the researcher. College authorities were not involved at any stage. These students used their personal mobile phones for food data recoding. After 21 days of food data capture, these students visited the lab where the researcher saved the pictures to a computer.

At the beginning and end of the study, height and weight were measured using a weighing scale and tape measure at the school health center/laboratory. At times when the participants could not or forgot to take a picture of the food, they sent messages to the researcher mentioning food items and the approximate time of ingestion. The researcher was in continuous yet random contact with volunteers to remind them to take pictures. The timing of ingestion for each participant data was examined for reporting compliance. Data were considered complete if the user logged at least 3 items every day (compliant day) and had 14 compliant days within 21 days of data collection.

### Calorie estimation from food pictures

After 21 days, the pictures from their cameras were downloaded and analyzed for time stamps and food pictures. All participants were allocated a unique ID for anonymity, in accordance with the ICMR guidelines of India. Picture details were manually annotated using a database from CalorieKing, MyFitnessPal websites and the Food and Nutrient Database for Dietary Studies (FNDDS) website of the USDA National Nutrient Database for Standard Reference to assess the caloric values of food items. In essence, food item name and portion size were visually assessed and cross-checked with a follow-up questionnaire [11]. It is important to reiterate that in India, especially for home-cooked food, the portion sizes often include multiple servings. Thus, the pictures taken within 5 minutes were treated as a single meal. A single picture with less food was standardized with a student's self-reported information; for example, if a student reported that his normal diet comprised two chapattis, then even if his picture(s) showed one chapatti in the meal(s), we considered the volume to be equivalent to two chapattis.

### Assessment of quality, quantity and timing of ingestion

Analyses of all pictures led to 173 distinct food/beverage items in the present study. These items were broadly grouped into water, FF or processed food, and natural or prepared food categories, depending on criteria such as commercial availability, caloric size per 100 grams and ease of access.

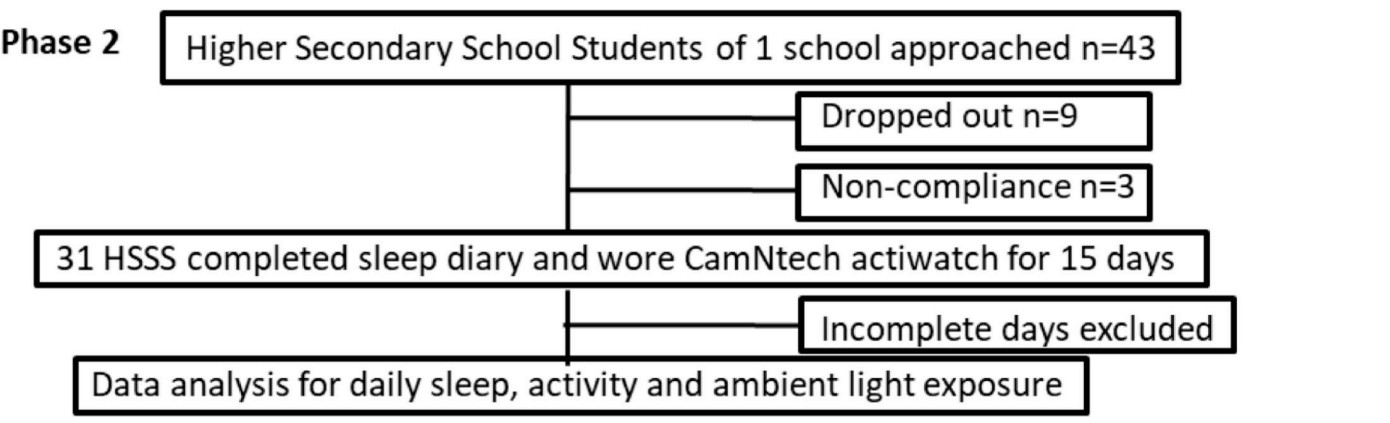

**Fig 1. Flow diagram showing student recruitment in two phases of the study.**

A total of 102 food items were categorized as high caloric/energy dense/FF. The caloric content per 100 grams of traditional food (all nuts, fresh fruits, pulses, vegetables, chapatti, white rice, milk, tea, etc.) ranged from 2.00 to 270 Kcal, with an average of 143.6 Kcal. The calorie content of 100 grams of different FF items (chips, desserts, pizzas, burgers, and snacks (including cream biscuits, stuffed

Indian paratha, fried foods, etc.)) ranged from 22.5 to 750 Kcal, with an average of 276.7 Kcal. The listed items were added to individual meals as per portion size assessed. The reported daily caloric intake for each participant was calculated from their food pictures on compliant days. The resting energy expenditure for each individual was calculated using the modified Harris-Benedict equation [35]. The data after quantitation were transferred to Graph Pad Prism software (version 7.0), San Diego, CA, USA, for representation and statistical analysis. We used t-tests and one-way ANOVA as applicable and considered p-values < 0.05 as significant. The timing of first to last caloric intake during the day was used to calculate the daily eating time window or eating duration, within which a person had a 95% probability of eating (between 2.5 and 97.5 percentile interval of all time-stamps in a day). For food data, a metabolic day was considered to begin at 4.00 a.m., such that the feeding duration of students ranged from 04:00 to 03:59 a.m. the next day (or 04–28 hours) [11]. All events at the time of eating reported by all participants were pooled, and the frequency distribution of eating events within a one-hour bin over 24 hours of the day was derived. Students who frequently forgot to take food pictures were excluded.

We aimed to track daily eating patterns, assess food habits and test the inclination of adolescents towards high caloric FF through schooling transitions with age.

## Sleep, activity and light exposure patterns among adolescent students

**Inclusion and exclusion criteria for actigraphy study.** The study protocol was approved by the HEC, MMH College, Ghaziabad, India. Data were collected from students of one public school in the Delhi area. The inclusion criteria were as follows: (1) students were enrolled in class XII, and (2) they agreed to wear motion-watches (CamNtech motion-watch 8, Motion-watch™) on their nondominant hand for 15 days. Exclusion criteria were as follows: (1) frequently forgot to wear the watch, (2) had a swimming routine, (3) discouraged by class teacher for reasons such as discipline, (4) undertook travel during the study, (5) had special needs, and (6) was not interested in the study.

In study phase 2, 43 students were approached to wear the motion-watch. Of these students, 12 dropped out and 31 contributed to the study (Fig 1). Motion-watches were initialized with a 1-minute epoch interval. Participants were asked to wear the motion-watch at all times (except when bathing) and to press "event marker" button when getting in and out of bed. A diary was provided to document unusual activities during the day that might impact the recording (e.g., device removal during sports or gym activities). Days with >4 hours of missing data were excluded. In certain cases, for a <1-hour duration, missing activity values were imputed on the basis of subject-specific average over all of the recording days at the same time period. Data of the total activity in 1440 bins (in 1-minute epochs) from 00:00 hours of day 1 to 11:59 hours of day 1 were averaged into 48 half-hour bins and summed up to the total activity in a day. Unlike a food study, in which the metabolic day is considered to start at 04.00 hours, sleep-wake data are described using 00:00 a.m. as 0 hours. An eight-hour window between 22:00 hours and 06:00 hours was considered night-sleep time. Activity/light during this period indicated "awake" status if the activity exceeded 499 MW counts per minute and the light >80 lux. The data of each volunteer plotted in 48 half-hour bins per day were normalized to account for interdevice variability. Due to the lack of clear onsets, offsets and fragmented patterns in quantified rest-activity circadian rhythms, data were analyzed for NPCRA.

## Results

### Challenges in collecting longitudinal eating pattern data

In contrast to adults, the collection of objective food consumption data from adolescents through smartphones was challenging with respect to schools, families and students. Many

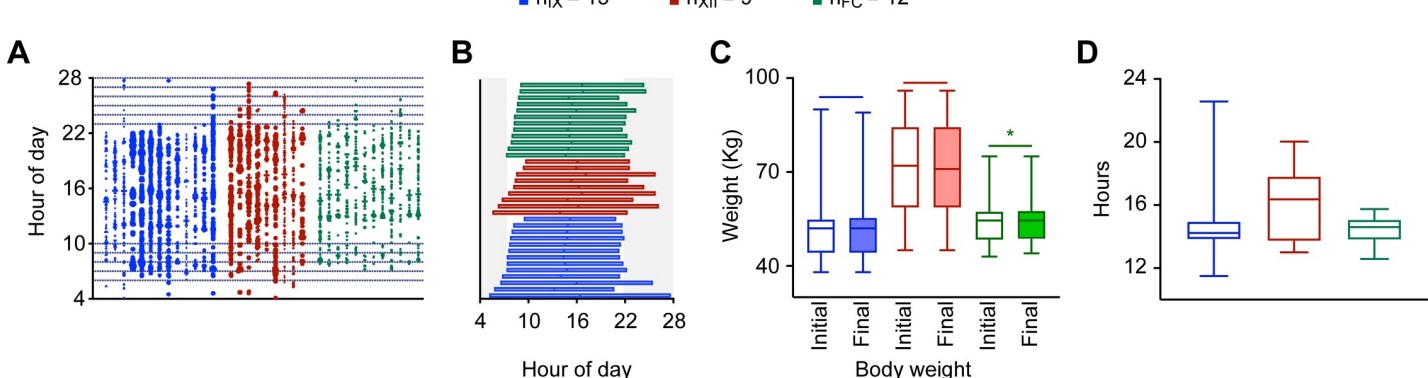

**Fig 2. Daily food eating patterns collected over 21 days for 34 students in high school (blue, $n_{IX}$ = 13), higher secondary school (red, $n_{XII}$ = 09) and first year of college (green, $n_{FC}$ = 12).** A. Scatterplot of all nonwater time-stamped ingestion events during the study, where a single vertical array represents one subject; B. Eating duration of individual participants arranged with increasing time of first meal; C. Initial (open boxes) and final (closed boxes) body weight of students (mean+25%ile and max-min range, paired t-test p<0.05) at the time of food recording; and (D) Groupwise variation in duration between first and last event of caloric ingestion (median+25%ile and min-max range).

parents were reluctant to allow their children to participate. In some cases, even after the parents consented, some of them associated their child's adverse performance at school with participation in the study and requested that their child be removed from the study. Students also frequently dropped out of the study due to classwork or tests. Some students and parents also felt that recording all food items was too distracting, and hence, they dropped out.

## Daily eating patterns and body weight differences among high school, higher secondary school and first-year college students

Overall, 34 students reported 651 days of data out of 714 potential days of recording (34 volunteers*21 days). A total of 6257 pictures and 102 messages were collected during those 651 days. Of the overall food/water events reported, 1166 were water events, 2409 were single items of food/beverage, and 2784 were pictures with multiple items (Fig 2A and 2B). The number of pictures reported daily per volunteer ranged from 6–18. Fig 2C shows the initial and final body weights of students in the 3-week food data capturing study. Although high school and HSSS did not show a significant change in body weight during the study, FY college students showed slight weight gain ($t_{12}$ = 2.436; P = 0.0331, two-tailed paired t-test) (Table 1).

**Table 1. Characteristics of the student cohorts in high school, higher secondary school and their first year of college.**

|  | High School | Higher Secondary School | First-year college |
|---|---|---|---|
| n (female,male) | 13 (7,4) | 9(6,3) | 12(9,3) |
| Age | 14.5 | 17.4 | 19.1 |
| Height | 155.4 | 165.7 | 157.5 |
| Initial BMI | 21.6 (16.6–36.9) | 26 (15.0–32.4) | 22.8 (15.5–33.92) |
| Final BMI | 21.6 (16.6–36.5) | 25.9 (15.03–32.4) | 22.9 (15.49–33.92) |
| Change in BMI | -0.013 | 0.04 | -0.12 |
| Paired t-test value | p = 0.75; $t_{12}$ = 0.3216 | p = 0.3466; $t_8$ = 1 | p = 0.0331; $t_{11}$ = 2.436 |
| Calculated REE (Kcal) | 1427.9 (1249–1982) | 1630.1 (1327–2156) | 1417.8 (1268–1869) |
| Average daily caloric intake reported (Kcal) | 1639.2 (1381–1909) | 1624.4 (1183.-1981) | 1753.3 (1491–2098) |
| percentage of REE reported | 116.2 (80.8–135.2) | 100.4 (78.1–131.7) | 124.5 (100.8–148.9) |

Changes in BMI are based on anthropometric observations and resting energy expenditure as calculated from self-reported food pictures.

Additionally, HSSS had significantly higher body weight (Kruskal-Wallis test, p<0.05) than did the other two groups.

## Timing of daily caloric intake

Fig 2A shows a scatterplot to illustrate the time of intake of nonwater food/beverage items by students on a 24-hour scale, i.e., from 04.00 h to 03.59 h. From Fig 2A, the eating duration was calculated as shown in Fig 2B. Eating duration ranged from 11.5 to 22.5 hours, with the median duration for students in high school, HSSS and FY of college being 14.23 hours, 16.36 hours and 14.6 hours, respectively (Fig 2D). Although the three-meals-daily pattern of eating was largely absent in all groups, there were larger variations in eating durations among HSSS (Fig 2D).

Intergroup differences were observed in the hourly distribution of the total number of caloric ingestion events (Fig 3A, 3B and 3C) and the hourly distribution of total calories consumed (Fig 3D, 3E and 3F). High school students exhibited frequent peaks in hours of eating. These peaks were observed before school started, during snack breaks, after school and during dinner. However, these spikes in food intake were missing in HSSS and FY college students. The time gap between two consecutive meals also varied among groups (Fig 3G, 3H and 3I), such that the median time gap decreased from 3.2 hours in high school students to 1.9 hours in college students.

Eating times among high school, HSSS and FY college students differed significantly ($F_{2, 5385}$ = 7.57, p<0.001) on weekdays and weekends ($F_{2, 823}$ = 5.15, p<0.001). When eating times on weekdays and weekends were compared within group(s), they did not differ in high school students ($t_{2925}$ = 1.098, p = 0.272) but did differ among HSSS ($t_{1726}$ = 2.291, p<0.05, student's two-tailed t-test) and FY college students ($t_{1557}$ = 2.265, p<0.05, student's two-tailed t-test). Students exhibited a delay of 22–26 minutes at the first meal time and a delay of 54–56 minutes at the last meal time on weekends compared to on weekdays (Fig 4B).

## Quality of daily caloric intake

We considered hourly intervals with <1% of total daily calorie intake as likely fasting or low-calorie hours. There was an interesting trend in the low-calorie hours. The low-caloric hours spanned 11 p.m. and 5 a.m. in high school students and 12 a.m. to 7 a.m. in college students, but they were alarmingly reduced to 2 a.m. to 4 a.m. in HSSS. We also examined the temporal pattern of FF consumption. The hourly distribution of total caloric and high-caloric food consumption (Fig 3D, 3E and 3F) was different in high school students than in higher secondary school or first-year college students. The times when FF% exceeded 40% of total hourly caloric intake differed, pointing towards differences in access to FF. The average number of FF intake events among school students was 30.5% of total caloric intake, while it was 35.6% in college students (Fig 4A, boxplots). As hypothesized, the percent caloric contribution of FF to total caloric intake was much higher (35–36% in school students and 44.5% in college students; Fig 4A, dot plots).

## Quantity of daily caloric intake

Of the 2408 food items consumed by high school students, 746 were FF, amounting to 30.5% of fast food (FF%); similarly, of the 39,497 Kcal consumed, 16,516 Kcal were FF, amounting to 35.42% fast food calories (FFc%). Similarly, figures for FF% in higher secondary school and FY college students were 30.5% and 35.6%, respectively, and those for FFc% were 36.16% and 44.5%, respectively. Despite the high FFc% among FY college students, the correlation between FF% and BMI was not significant (r = -0.3357, P = 0.1434), whereas there was a negative (r =

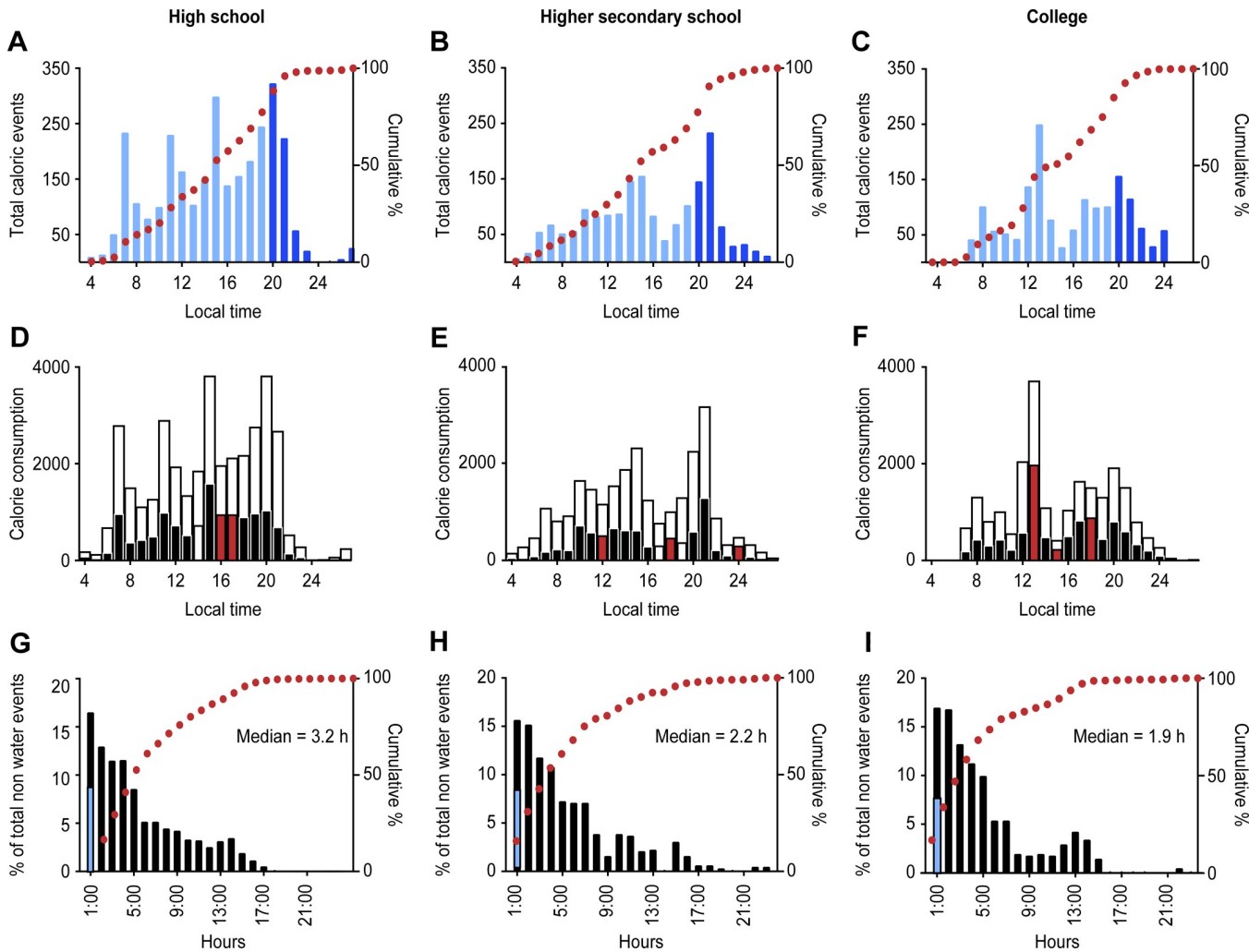

**Fig 3. Temporal details of daily food consumption data collected over 21 days for 34 school students in high school (left panel), higher secondary school (middle panel) and first year of college (right panel).** A-C. Intergroup differences were observed among high (A), higher secondary (B) and FY college (C) in the hourly distribution of the number of nonwater ingestion events; the conventional three-meals-a-day pattern was not found in any group. The dotted curve shows the cumulative number of ingestion events in a day. D-F. Hourly distribution of total calories consumed (open bars), calories consumed as fast food (closed bars) and times when FF% exceeded 40% of total hourly caloric intake (red bars), G-I. Frequency distribution of intervals between consecutive caloric ingestion events. FF events contributing to <1 h intermittent gap, are shown in Cyan blue area. Group differences in the median duration of inter-meal intervals existed, and only 10% of events had an inter-meal interval of >10 hours in all groups.

-0.7033, P< 0.01) and positive (r = 0.7667, P< 0.05) significant correlation between FF% and BMI in high school and HSSS, respectively (Fig 4C). FF% does not appear to be related to the daily duration of eating in all three groups (Fig 4D).

## Sleep and activity patterns among adolescent students

Sleep, activity and light exposure differed among 31 HSSS. On the basis of differences in sleep patterns (Fig 5A), we classified them into four groups: regular sleepers (RS, n = 10), who slept at least 6 hours every night (Fig 5B); short sleepers (SS, n = 9), who slept every night but only for 2–5 hours (Fig 5C); irregular sleepers (IrS, n = 5), who frequently skipped night sleep (Fig 5D); and nonsleepers (NS, n = 7) who did not exhibit sleep between 22:00 hours and 06:00 hours (Fig 5E). Thirteen of the 15 HSSS volunteers were contacted after the completion of

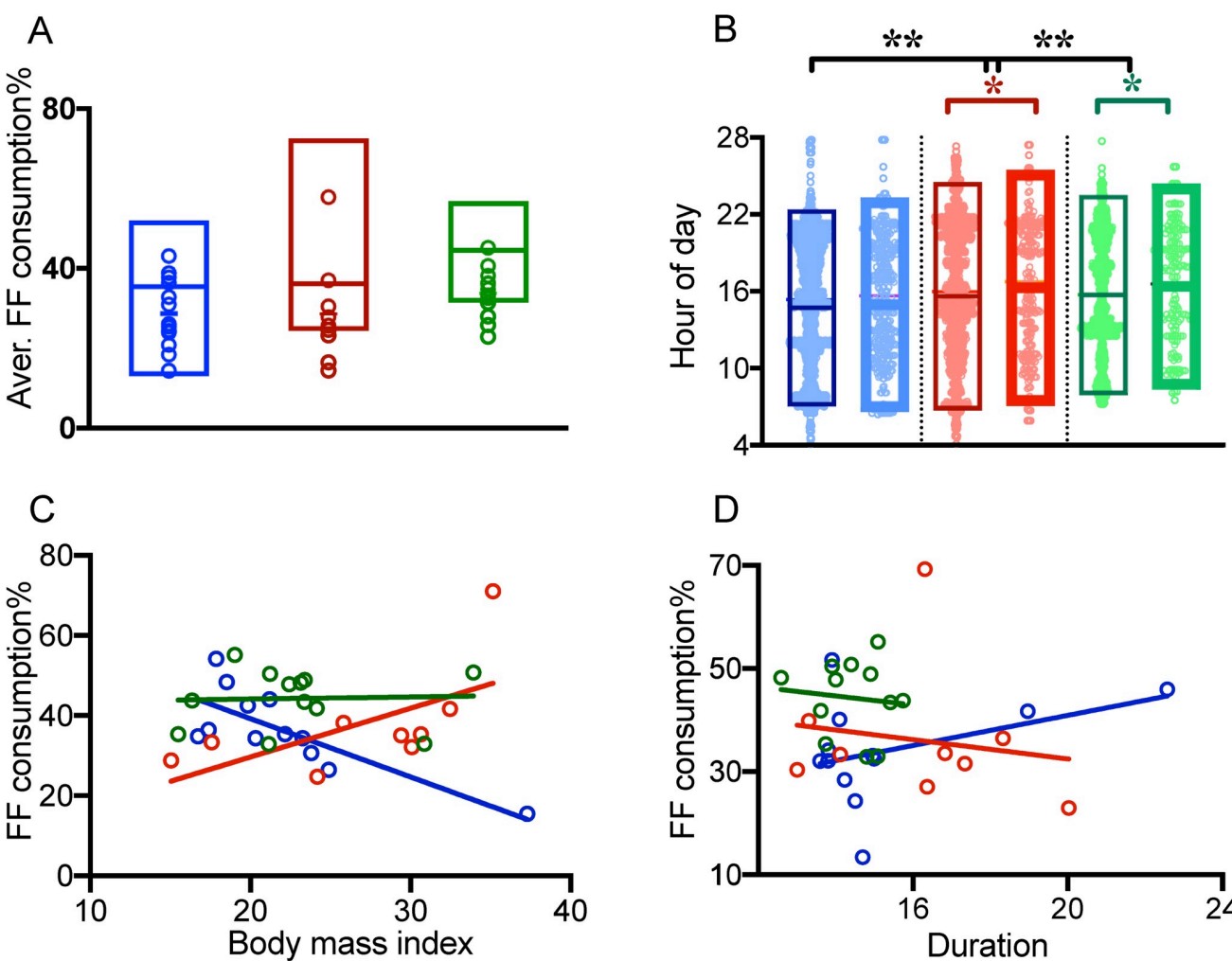

**Fig 4. Fast food (FF) consumption supplements circadian disruption in daily food eating patterns.** Parameters of caloric ingestion events containing FF were separately analyzed from the food picture data of 34 school students in high school (blue), higher secondary school (red) and their first year of college (green). A. The aligned dot plot shows the average percentage of events containing FF from total nonwater ingestion events; the box plot shows that the percent caloric contribution of FF to total caloric intake was higher B. Group-wise scatter dot plot of FF events on weekdays (left box of column) and Sundays (right box of column) to show that FF is eaten randomly at any time of day. On Sundays (thick boxes, min-max), the eating duration was slightly delayed in all three groups. C. The percentage of FF consumed was directly proportional to the BMI of higher secondary school students. The converse was true for high school students, whereas no correlation was found between the percentage of FF consumed and the BMI of college students. D. The percentage of FF consumed was positively correlated with the total duration of daily food consumption in high school students compared to the other two groups.

actigraphy to provide reasons for their diminished night sleep. These HSSS confirmed that they studied at night. All HSSS except 6 regular and 3 short sleepers exhibited daytime napping/sleeping. Furthermore, to evaluate differences in the quality of night sleep, we compared nighttime activity (22.00 hours– 06.00 hours) using sleep groups as the between-subject factor and the day of the week (weekday vs. weekend) as the within-subject factor. Repeated measure one-way ANOVA revealed significant differences in night sleep on weekends (p<0.0001, $F_{15,45} = 38.58$) and weekdays (p<0.001, $F_{15,45} = 24.05$). Apart from weekly differences in sleep patterns, the daily activity of the four groups differed significantly with respect to time of day (repeated measures ANOVA, $F_{46, 9823} = 15.8$, $p< 0.001$).

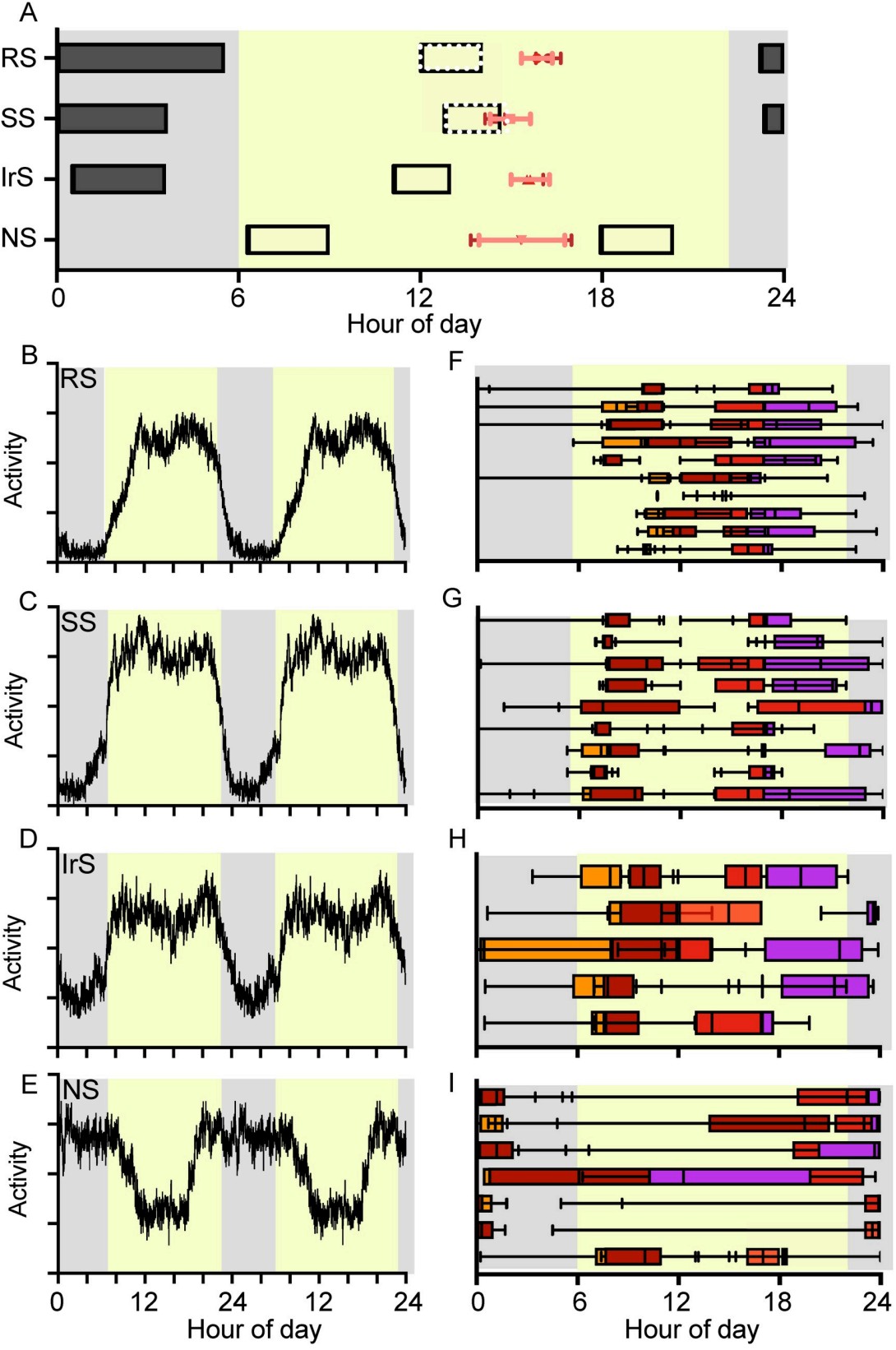

**Fig 5. Objective measures of daily sleep/physical activity and ambient light exposure.** Thirty-one higher secondary school students (HSSS) continuously wore the CamNtech motion-watch on their nondominant wrist for up to 15 days. All graphs are shown on the 24-hour x-axis, with a light yellow area in the background representing daytime (from 06.00 hours to 22.00 hours) and a gray area representing the night phase. A. Average daily transition dynamics of sleep (horizontal bars) and acrophase of activity (dots). Daily activity in MW counts was quantified in 1-minute epochs and summed into half-hour bins. The peak (acrophase) activity timings of HSSS could be compared for weekday (deep red) and weekends (light red). The students were grouped according to their night sleep timings shown by closed horizontal bars i.e., regular RS, short SS, irregular Irs and non-NS sleepers. Some students exhibited daytime sleep (open horizontal bars), and the groups in which only a few students slept during the day are shown by dotted bar outlines. B-E. Double-plotted activity profiles of RS, SS, IrS and NS. RS exhibited day activity and a > = 6-hour night of sleep; SS exhibited day activity and a 2- to 5-hour night of sleep, Irs showed day activity often alternated with night activity, and NS did not exhibit sleep during the night. F-I. Corresponding luminance data of RS, SS, Irs and NS groups to identify times of exposure to light intensity using the photometric measure of the motion-watch. Each horizontal bar represents one student (orange-first 100 lux; deep red-first 500 lux; red-last 500 lux; purple- last 500 lux).

Overall, the daily individual activity patterns of 31 HSSS were consolidated into 19.8± 0.32 hours in a 24-hour span (Fig 4B–4D). In three groups, i.e., regular, short and irregular sleepers,

**Table 2. Nonparametric circadian rhythm assessment (NPCRA) of 31 higher secondary school students for select days (shown in parentheses).** The following values are provided in columns- RA: relative amplitude, IS: interdaily stability, and IV: intradaily variability and night activity as a percentage of total daily activity.

| Volunteer (days) | RA | IS | V | % activity (night) Mean (SE) |
|---|---|---|---|---|
| A (5) | 0.828 | 0.567 | 0.858 | 0.45 (0.3) |
| B (3) | 0.921 | 0.649 | 0.902 | 0.41 (0.1) |
| C (6) | 0.981 | 0.752 | 0.665 | 1.72 (0.9) |
| D (3) | 0.862 | 0.73 | 0.654 | 0.73 (0.3) |
| E (13) | 0.925 | 0.372 | 1.031 | 1.32 (0.5) |
| F (13) | 0.478 | 0.158 | 0.93 | 3.21 (3.2) |
| G (13) | 0.961 | 0.617 | 0.736 | 5.25 (2.4) |
| H (16) | 0.899 | 0.547 | 0.573 | 2.91 (0.7) |
| I (14) | 0.932 | 0.595 | 0.843 | 1.89 (0.3) |
| J (9) | 0.999 | 0.386 | 0.697 | 1.55 (0.4) |
| K (6) | 0.854 | 0.603 | 0.869 | 1.39 (0.2) |
| L (6) | 0.883 | 0.551 | 0.774 | 5.63 (4.6) |
| M (7) | 0.972 | 0.576 | 0.946 | 1.93 (0.7) |
| N (11) | 0.93 | 0.613 | 0.873 | 8.19 (7.1) |
| O (14) | 0.902 | 0.525 | 0.741 | 2.97 (1.5) |
| P (10) | 0.879 | 0.539 | 1.055 | 0.89 (0.3) |
| Q (16) | 0.857 | 0.406 | 1.305 | 6.25 (6.2) |
| R (8) | 0.175 | 0.109 | 1.105 | 4 (1.7) |
| S (16) | 0.927 | 0.603 | 0.958 | 0.59 (0.3) |
| T (15) | 0.919 | 0.514 | 0.794 | 0.85 (0.2) |
| U (4) | 0.934 | 0.496 | 0.65 | 4.63 (1.2) |
| V (3) | 0.914 | 0.668 | 1.047 | 1.24 (0.4) |
| W (4) | 0.894 | 0.435 | 0.764 | 2.1 (0.4) |
| X (15) | 0.812 | 0.416 | 0.843 | 2.13 (0.5) |
| Y (6) | 1 | 0.615 | 0.579 | 33.7 (8.5) |
| Z (8) | 0.849 | 0.436 | 0.515 | 30 (8.6) |
| AA (6) | 0.942 | 0.707 | 0.879 | 22.5 (7) |
| AB (14) | 0.899 | 0.447 | 0.987 | 22.3 (9.8) |
| AC (5) | 0.944 | 0.602 | 1.078 | 26.8 (6.7) |
| AD (13) | 0.938 | 0.697 | 0.717 | 38.7 (2.9) |
| AE (19) | 0.669 | 0.361 | 0.907 | 10.3 (2.5) |

there was no pattern of morning activity onsets, but late evening offset ranged from 11.30 p.m. to 01.00 a.m. Among these students, at least 2–3 nights (between 10.00 p.m. and 6 a.m. next day) on weekdays (Fig 5A) every week were characterized by moderate to high levels of activity. There was no visible trend of activity onsets or offsets in nonsleepers.

### Light exposure pattern among adolescent students

Within individual variations in light exposure, there was agreement with respect to low IS scores (Fig 5F, 5G, 5H and 5I). The RS and SS HSSS (Fig 5B and 5C) received light exposure to the first 500 lux or above during the day (Fig 5F and 5G), while IrS and NS received light exposure to the first 500 lux or above during the night (Fig 5H and 5I). Notably, the 'environmental' night was the most active and bright phase of day for 22% of the cohort (Fig 5I).

NPCRA revealed peak/trough changes through relative amplitudes ranging from 0.18 to 0.83, indicating sedentary to active lifestyles of students. The amplitude, consolidation, stability, and percentage of night activity also varied; the nonsleeper HSSS exhibited 1/3 of daily activity at night (Table 2). Irregularities in daily routines were quantified as high consolidation (IV- 0.51–1.3) and lower stability (IS- 0.1–0.75). Such high IV or lower IS indicated rest during the day/activity at night, thus representing circadian rhythm disruption (CRD). RA and IS were significantly correlated (r = 0.75; p<0.0001), meaning that less activity in HSSS was related to irregularity in daily activity/rest routines. IS was directly related to rhythm amplitude and light exposure and was high only when there was less night activity and better photic synchronization, indicating quality life measures.

## Discussion

The increasing concern for the rise of lifestyle-related issues in adolescents is largely based on data projections of questionnaire-based surveys and/or diary recall. First, these methods lack evidence, and second, they are highly subjective, with retrospective influence on volunteer reporting. A smartphone capture-based approach [9] minimizes such glitches in human longitudinal studies involving eating behavior (Table 1). Notably, despite the better availability of relatively inexpensive data plans in India to date, the present study employed inexpensive camera phones instead of apps/fully functional smartphones to collect nutrition data to satisfy school regulations, especially for high and higher secondary school students. In food eating pattern studies, the possibility of false negatives arising from an inability to report or forgetting to report cannot be ruled out. The body weight of the students was also not affected by the acquisition of nutrition data. Nevertheless, actual energy intake for school students was higher than the resting energy expenditure (REE) calculated, ruling out an adverse effect of capturing food data on daily food intake.

Despite the small cohort size, there were clear differences in duration (time from first caloric ingestion to last meal of day), frequency, quality and caloric quantity of daily food intake among high school, higher secondary school and FY college students. High school students and FY college students exhibited a median duration of eating of ~14 hours, which was less than that of HSSS. The latter exhibited a 16.36-hour daily duration of eating. Although maximum FF% was observed in the FY college students' diets, HSSS had a higher BMI, indicating a negative effect of a longer eating duration, i.e. reduced night fasting. The qualitative assessment of food consumption also supported the adverse impact of reduced night fasting in HSSS, such that they exhibited a positive correlation between FF% and BMI. This finding is consistent with an earlier view suggesting that eating later in the day enhances the chances of obesity [16, 36]. The differences in eating pattern became conspicuous when plotted in hourly bins (Fig 3A, 3B and 3C). High school students exhibited four different times of peak eating events, with

the largest peak at 21–22 hours (Fig 3A and 3D), which suggests a postprandial assimilation of food at night and, hence, night hyperglycemia [37]. Additionally, these adolescents have school starting at 06.00/07.00 a.m. in the morning, as evident from the first (morning) peak of ingestion events. Inadvertently, night fasting seldom exceeded 9 hours, since it takes up to 2 hours to assimilate food, which implied up to 7 hours of rest to the liver. Even >50% of inter-meal gaps are less than 5 hours (Fig 3G). Such reduced night fasting is a risk factor for metabolic syndrome [38].

We expected a conventional three-meals-a-day pattern of eating among school students. However, any such daily pattern was largely absent. High school students exhibited a higher number of ingestion events in the morning, before school, during tiffin break, after school and at dinner time. They also exhibited more than a 60-minute delay in the first ingestion event on the weekends. There was larger variation in daily eating duration in HSSS than in high school and FY college students (Fig 2B). An earlier study on the eating patterns of Indian urban adults revealed larger variation in dinner than in breakfast times [11]. The present observations for students were similar to those for adults, as these students also exhibited larger variation in the last meal of the day. Weekday-weekend differences in the duration of eating were significant among HSSS and FY college students. Metabolic homeostasis misaligns with the mistiming of food, altering glucose and energy metabolism [33].

The daily eating patterns in hourly bins did not exhibit clear daytime peaks in HSSS, indicating disruption of daily rhythms of eating (Fig 3B and 3D). A number of caloric intake events were spread throughout the day, less than half of which occurred before 05.00 p.m. Herein again, approximately 50% of inter-meal gaps were less than 5 h (Fig 3H). The lack of lunch/ dinner peaks indicates meal irregularity. Such erratic eating patterns are associated with a reduction in the thermic effect of food and higher glucose responses, thus reducing metabolic health [39]. In the present study, we observed not only greater meal irregularity among HSSS but also larger meal sizes (i.e., greater FFc%), which were positively correlated with an increase in BMI, thus increasing the possibility of long-term weight gain [40]. The FY college students exhibited nearly three inconspicuous peaks in the number of ingestion events. Additionally, the number of events per volunteer was lower by at least 10–15% compared to those for school students (Fig 3C and 3F). The college students' data revealed that 1) the first ingestion event was ~2 hours later than that of school students and 2) they had greater ease of access to FF during the day. Unlike school students and more similar to adults [11], approximately 40% of ingestion events occurred among college students by noon. Although they exhibited better night fasting than school students, half of the inter-meal gaps occurred in less than 5 hours (Fig 3I). An increase in the energy-dense diet characterized the food quality of college students.

Dietary quality was an important parameter studied. FF comprised 2/5 of the total caloric intake. It has been widely investigated that mice randomly eating a high-fat diet tend to develop metabolic diseases faster than those eating a normal diet (16, 17). This is because a high-fat, energy-dense diet, along with disruptions in eating patterns, overrides the circadian clock. Energy-dense diets have pleiotropic effects that lead to the reprogramming of the metabolic and transcriptional liver pathways. Many oscillating transcripts and metabolites are phase-advanced by fast food, disrupting the circadian clock [41]. It cannot be ruled out that the lack of a positive relationship between FF% and daily duration of eating (Fig 4D) in the present study might have resulted from the small cohort size and needs to be further investigated.

We observed wakeful activities far into the night among HSSS. Owens [42] reviewed the sleep patterns in American adolescents, factors contributing to chronic sleep loss, and reported that sleep impairments are an important public health issue. Sleep restriction is a serious threat

to the academic success and safety of adolescents, resulting in health-related consequences, such as depression and increased obesity risk. The actimetry data in HSSS also confirmed the reduced availability of sleep hours. A late-night sleep onset ranging from 11.30 p.m. to 01.00 a.m. in 3/4 of the cohort was alarming [43]. In another study, the influence of the sleep patterns of 2,259 adolescent students was examined using latent growth cross-domain models. It was reported that students who obtained less sleep exhibited lower initial self-esteem and higher initial levels of depressive symptoms [44].

The strength of this study is evidence-driven data capturing all subjects within a narrow age and socioeconomic range. Disruption in the daily food rhythms of adolescents indicated an intricate relationship with disrupted sleep-activity rhythms during follow-up. The measurement and comparison of sleep, activity level and light exposure in Indian high school students has never been performed before.

Our feasibility study on school students has many limitations. We have not measured circadian phase markers such as urinary melatonin, so we missed a circadian misalignment interpretation of some important observations, such as 'no night sleep', in nonsleeper HSSS. Although stress could have affected volunteer sleep characteristics, we did not include students' self-reported mood or stress because stress hormone(s) were not measured. Additionally, the cohort size was small in the food eating pattern study and sleep/light monitoring. Recruiting students from public schools to match the socioeconomic status of students was a precaution taken to minimize the study biases. Public school students mainly hail from the middle-income group, or the part of society focused on "education for employment". Therefore, in India, similar to a few other countries, the self-worth of a student in the middle-income group is determined by academic success and grades, thus increasing students' pressure to learn. India is top rated among suicides resulting from class XII exam failures [45]. Our study highlights intricate lifestyle issues that are detrimental to adolescent health, and the present study increases the scope for corrective interventions. Logan and McClung [46] recently proposed that brain disorders and circadian dysfunction are correlational, and conditional interventions, such as morning bright light therapy and better sleep hygiene during adolescence, can help reduce CNS disorders.

In addition to a baseline for future intervention studies in school students, the present study highlights circadian disruptions in feeding-fasting and activity-rest cycles in Indian school students and should be circulated for outreach and awareness purposes.

## Acknowledgments

The authors thank the volunteers for their anonymous contributions and Prof. Satchidananda Panda of the SALK Institute of Biological Sciences, USA, for suggestions during the preparation of the manuscript.

## Author Contributions

**Conceptualization:** Neelu Jain Gupta.

**Data curation:** Akansha Khare.

**Formal analysis:** Neelu Jain Gupta.

**Investigation:** Neelu Jain Gupta.

**Methodology:** Akansha Khare.

**Project administration:** Neelu Jain Gupta.

**Resources:** Neelu Jain Gupta.

**Software:** Neelu Jain Gupta.

**Supervision:** Neelu Jain Gupta.

**Validation:** Neelu Jain Gupta.

**Visualization:** Neelu Jain Gupta.

**Writing – original draft:** Neelu Jain Gupta.

**Writing – review & editing:** Neelu Jain Gupta.

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
