## [Decision Letter · Decision Letter 0]

18 Nov 2019

PONE-D-19-22701

Feasibility study shows disruption in daily eating-fasting and activity-rest cycles in Indian adolescents attending school

PLOS ONE

Dear Dr. Jain Gupta,

Thank you for submitting your manuscript to PLOS ONE. After careful consideration, we feel that it has merit but does not fully meet PLOS ONE’s publication criteria as it currently stands. Therefore, we invite you to submit a revised version of the manuscript that addresses the points raised during the review process.

We would appreciate receiving your revised manuscript by Jan 02 2020 11:59PM. To enhance the reproducibility of your results, we recommend that if applicable you deposit your laboratory protocols in protocols.io, where a protocol can be assigned its own identifier (DOI) such that it can be cited independently in the future. For instructions see: http://journals.plos.org/plosone/s/submission-guidelines#loc-laboratory-protocols

We look forward to receiving your revised manuscript.

Kind regards,

Etienne Challet

Academic Editor

PLOS ONE

Journal Requirements:

"This work was not supported by any external funding source. However, CamNtech motion

watches for data collection were given to NJG by Prof. Satchidananda Panda, SALK Institute

of Biological Sciences, USA.".

"The author(s) received no specific funding for this work".

5.  Please include your tables as part of your main manuscript and remove the individual files. Please note that supplementary tables (should remain/ be uploaded) as separate "supporting information" files

Additional Editor Comments (if provided):

In addition, I would suggest omitting "Feasibility study shows" in the title. This shortening would provide a more straightforward message.

Reviewers' comments:

Reviewer's Responses to Questions

**Comments to the Author**

1. Is the manuscript technically sound, and do the data support the conclusions?

Reviewer #1: Yes

Reviewer #2: Yes

2. Has the statistical analysis been performed appropriately and rigorously? 

Reviewer #1: Yes

Reviewer #2: Yes

3. Have the authors made all data underlying the findings in their manuscript fully available?

Reviewer #1: Yes

Reviewer #2: Yes

4. Is the manuscript presented in an intelligible fashion and written in standard English?

Reviewer #1: Yes

Reviewer #2: Yes

5. Review Comments to the Author

Reviewer #1: Despite of the limited cohort size, the results show concerning data about rhythmic misalignment in adolescents. There are not a lot of studies focusing on this group. Those information should be important in developing public policies to reduce this problem.

Reviewer #2: The present manuscript has provided an insight for the daily lifestyles of young students and their feeding-fasting and rest activity cycles. I have few minor comments and questions regarding the present manuscript:

Minor comments:

1. The presents study did not explains about the male and female students in all the groups of the study? Please justify the number of male and females in each group.

2. And also provide the caloric differences between male and female group.

3. In figure 2C and 2D why there is high deviation in High school group weight and eating duration?

4. In result section under heading Daily eating patterns and body weight. You have written 651 days of recording. Is study was carried for approximately 2 years? If yes than it would be great if you can provide the change in caloric intake of each group in relation to seasonal changes.

5. In result section under heading Timing of daily caloric intake your paragraph second where you discuss about Fig 3. The labeling of figure 3 and results are not similar 3C is written twice in figure.

6. In result section under heading quantity of daily caloric intake why their is high FFc% among FY student? Do they have more preference to high sugar food or high fat food?

6. PLOS authors have the option to publish the peer review history of their article (what does this mean?). If published, this will include your full peer review and any attached files.

Reviewer #1: No

Reviewer #2: Yes: Satish kumar Sen

---

## [Author Response · Author response to Decision Letter 0]

4 Dec 2019

Responses to Reviewer's Questions

Reviewer #1: Despite of the limited cohort size, the results show concerning data about rhythmic misalignment in adolescents. There are not a lot of studies focusing on this group. Those information should be important in developing public policies to reduce this problem.

Reviewer #2: The present manuscript has provided an insight for the daily lifestyles of young students and their feeding-fasting and rest activity cycles. I have few minor comments and questions regarding the present manuscript:

1. The presents study did not explain about the male and female students in all the groups of the study? Please justify the number of male and females in each group.

***Thank you for pointing out. Added in table 1. 

2. And also provide the caloric differences between male and female group.

***Thank you for pointing out, and that there is no significant difference (small cohort) among males and females in any of the three groups.

3. In figure 2C and 2D why there is high deviation in High school group weight and eating duration?

***only one student of High school group exhibited last mealtime between 22 hrs-26 hrs (late night eating) among high school students, but could not be ignored due to small cohort size. Similarly, one obese High school volunteer might’ve affected BMI data. Our purpose for BMI graphing was to compare initial and final values and its correlation with other study variables.

4. In result section under heading Daily eating patterns and body weight. You have written 651 days of recording. Is study was carried for approximately 2 years? If yes than it would be great if you can provide the change in caloric intake of each group in relation to seasonal changes.

***Although the data collection spread over 3 years, the number 651 days indicates the potential days when data was recorded and included in the study. Normally food data capturing by students could not be conducted near summer vacation(May-July) or half yearly (sept-Oct) of final exam (Feb-March) time, therefore, seasonal changes need to be studied separately.

5. In result section under heading Timing of daily caloric intake your paragraph second where you discuss about Fig 3. 

***Corrected. Thank you.

The labelling of figure 3 and results are not similar 3C is written twice in figure.

*** Thank you for pointing out, the mismatch was due to labelling error in figure during copyedit. Corrected. Hope it is in order now.

6. In result section under heading quantity of daily caloric intake why their is high FFc% among FY student? Do they have more preference to high sugar food or high fat food?

***Yes. Thanks for this important observation; FY students consumed more fast-food, but this FF consumption was restricted to day-time hours (Fig. 3F).

---

## [Editor Report · Decision Letter 1]

11 Dec 2019

Disruption in daily eating-fasting and activity-rest cycles in Indian adolescents attending school

PONE-D-19-22701R1

Dear Dr. Jain Gupta,

We are pleased to inform you that your manuscript has been judged scientifically suitable for publication and will be formally accepted for publication once it complies with all outstanding technical requirements.

With kind regards,

Etienne Challet

Academic Editor

PLOS ONE
---

## [Editor Report · Acceptance letter]

17 Dec 2019

PONE-D-19-22701R1 

Disruption in daily eating-fasting and activity-rest cycles in Indian adolescents attending school 

Dear Dr. Jain Gupta:

I am pleased to inform you that your manuscript has been deemed suitable for publication in PLOS ONE. Congratulations! Your manuscript is now with our production department. 

With kind regards,

on behalf of

Dr. Etienne Challet 

Academic Editor

PLOS ONE